# The Switch and Reciprocating Models for the Function of ABC Multidrug Exporters: Perspectives on Recent Research

**DOI:** 10.3390/ijms24032624

**Published:** 2023-01-30

**Authors:** Peter M. Jones, Anthony M. George

**Affiliations:** School of Life Sciences, University of Technology, Sydney, NSW 2007, Australia

**Keywords:** ABC transporters, nucleotide binding domains, transmembrane domains, NMR and mass spectrometry, reciprocating model

## Abstract

ATP-binding cassette (ABC) transporters comprise a large superfamily of primary active transporters, which are integral membrane proteins that couple energy to the uphill vectorial transport of substrates across cellular membranes, with concomitant hydrolysis of ATP. ABC transporters are found in all living organisms, coordinating mostly import in prokaryotes and export in eukaryotes. Unlike the highly conserved nucleotide binding domains (NBDs), sequence conservation in the transmembrane domains (TMDs) is low, with their divergent nature likely reflecting a need to accommodate a wide range of substrate types in terms of mass and polarity. An explosion in high resolution structural analysis over the past decade and a half has produced a wealth of structural information for ABCs. Based on the structures, a general mechanism for ABC transporters has been proposed, known as the Switch or Alternating Access Model, which holds that the NBDs are widely separated, with the TMDs and NBDs together forming an intracellular-facing inverted “V” shape. Binding of two ATPs and the substrate to the inward-facing conformation induces a transition to an outward conformation. Despite this apparent progress, certainty around the transport mechanism for any given ABC remains elusive. How substrate binding and transport is coupled to ATP binding and hydrolysis is not known, and there is a large body of biochemical and biophysical data that is at odds with the widely separated NBDs being a functional physiological state. An alternative Constant Contact model has been proposed in which the two NBSs operate 180 degrees out of phase with respect to ATP hydrolysis, with the NBDs remaining in close proximity throughout the transport cycle and operating in an asymmetric allosteric manner. The two models are discussed in the light of recent nuclear magnetic resonance and hydrogen-deuterium exchange mass spectrometry analyses of three ABC exporters.

## 1. The Ubiquity of ABC Transporters: Architecture and Function

ATP-binding cassette (ABC) transporters comprise a large superfamily of primary active transporters, which are integral membrane proteins that couple energy to the uphill vectorial transport of substrates across cellular membranes, with concomitant hydrolysis of ATP. ABC transporters are found in all living organisms, coordinating mostly import in prokaryotes and export in eukaryotes, with exceptions that ferry allocrites in the other direction in each division [1,2,3,4,5,6,7]. ABC transporters are of great biomedical importance, being involved in physiological and pathological processes, including multidrug resistance exhibited by cancers, parasites, and pathogenic bacteria; stem cell differentiation; vision [4,8,9]; lipid and cholesterol metabolism [10]; and degenerative proteopathies [11,12].

ABC transporters typically comprise two transmembrane domains (TMDs) that form the channel or conduit for binding and translocating allocrites, and two nucleotide binding domains (NBDs) or ATP-binding cassettes, which hydrolyze ATP to energize the translocation of allocrites across the cell membrane (Figure 1A). The NBDs are highly conserved and unambiguously monophyletic. The NBDs contain sequence motifs involved in ATP binding and hydrolysis, namely the Walker A and B motifs common to P-loop ATPases [13], and a number of flexible loops, designated by conserved residues at their N- or C-termini, namely the D-, H-, Q-, and A-loops, as well as sequence motifs found in subgroups such as the X-loop in MDR exporters [4,14,15].

Unlike the highly conserved NBDs, sequence conservation in the TMDs is low, with their divergent nature likely reflecting a need to accommodate a wide range of substrate types in terms of mass and polarity [16,17,18].

Structurally, the NBD monomer is bilobal, comprising a RecA-like principal domain [19], which contains the Walker A (P-loop) and Walker B ATP-binding motifs, and a flexibly attached helical domain, which contains the “LSGGQ” signature or C-motif, diagnostic of the ABC-ATPase superfamily [1,20] (Figure 1B).

The NBD dimer structural configuration was first proposed by [21]. Subsequent crystal structures confirmed this NBD “sandwich” dimer with ATP binding sites at the intermonomer interface, opposed by the Walker A and B sequences of one monomer and the C-motif of the opposite monomer. These composite ATP-binding sites are referred to as nucleotide binding sites (NBSs) [21,22] (Figure 1C).

An explosion in high resolution structural analysis over the past decade and a half has produced a wealth of structural information for ABCs [23,24,25,26]. This has revealed that while the NBDs are structurally conserved, as expected from their primary sequences, for the TMDs, a number of different folds exist, and these have suggested distinct mechanisms. One commonality nonetheless is that in both importers and exporters, the TMDs and NBDs interact via short “coupling helices” (CHs) that form turns in intracytoplasmic loops (ICLs), parts of the TMDs that extend into the cytosol [6].

Notwithstanding their diversity, the TMD structures are generally construed to form a transmembrane conduit that alternates between inward- and outward-facing conformations. Thereby, the putative channel is, in most cases, accessible from only one or other side of the membrane during a transport cycle [7,25,26,27], in line with the original Jardetzky allosteric model [28].

Based on the structures, a general mechanism for ABC transporters has been proposed, which is known as the Switch or Alternating Access Model [29], illustrated in Figure 2. This mechanism was originally based on structures of the bacterial MDR exporter homologues Sav1866 [30] and MsbA [31], where two distinct conformations were observed: outward facing (OF), in which the NBDs form a closed dimer with one ATP or ATP analogue bound in each of the two active sites (NBD sandwich dimer), with the TMDs forming an extracellular-facing “V” shape, and inward facing (IF), in which the NBDs are widely separated, with the TMDs and NBDs together forming an intracellular-facing inverted “V” shape. Binding of two ATPs and the substrate to the IF conformation induces a transition to the OF conformation whereupon substrate release to the extracellular side and ATP hydrolysis induces the transition back to the IF to complete the export cycle.

Despite this apparent progress, however, clarity and certainty around the transport mechanism for any given ABC transporter remains elusive. In particular, how substrate binding and transport is coupled to ATP binding and hydrolysis is not known, and there remains significant uncertainty around the operation of NBDs. Thus, while in many essential respects, the Switch mechanism is a necessary implication of the structures, NBD dimer dissociation to widely separated monomers is required for the substrate to enter and bind to the TMDs in exporters, for example, a large body of biochemical and biophysical data is at odds with the widely separated NBDs being a functional state [23].

Based on such data and MD simulations of ABC NBDs, we previously proposed a Constant Contact model for the operation of the NBDs [5]. In this scheme, the two NBSs operate 180 degrees out of phase with respect to the catalytic cycle of ATP hydrolysis, with the NBDs remaining in close proximity throughout the transport cycle and operating in an asymmetric manner.

To integrate substrate transport into the Constant Contact scheme, we proposed a Reciprocating Model (RM) for the integrated transport mechanism whereby substrate binding, translocation, and release in the TMDs is coupled directly with ATP binding, hydrolysis, and product release in the NBDs [35] (Figure 3). In this model, there are two equivalent substrate transmembrane translocation channels (TMCs) embodied by the TMDs, each coupled to one NBS; these two coupled TMC–NBS pairs operate 180 degrees out of phase (Figure 4). We showed how these ideas are consistent with a large body of data for ABC transporters [35].

Significantly, however, the RM appears unable to be reconciled with 3D structures of ABCs. Nonetheless, we recently argued that the current situation with respect to progress in understanding the ABC transporter mechanism is indistinguishable from one in which the structures deviate significantly from the physiological state. On that basis, we suggested that the structures ought to be regarded with a greater degree of circumspection, and not as the pre-eminent arbiter of all other data, as it appears is now generally the case.

Extending the debate from this perspective, here we review and discuss recent nuclear magnetic resonance (NMR) and hydrogen-deuterium exchange mass spectrometry (HDX) analyses of ABCB1 and two of its bacterial homologues, namely, mammalian P-glycoprotein, involved in xenobiotic resistance, and its bacterial homologues MsbA, a lipid A exporter involved in production of outer membrane liposaccaharides, and BmrA involved in antibiotic resistance in Bacillus subtilis. MsbA and BmrA are homodimers of protomers, which comprise one TMD fused to one NBD. P-glycoprotein is a single polypeptide comprising homologous halves each consisting of one TMD fused to one NBD. Shared structural features of these ABCB1 ABC exporters are illustrated in Figure 1.

## 2. Do Recent NMR and HDX Studies of ABC Exporters Depart from the Reciprocating Model?

Spadaccini et al. (2018) [36] used NMR techniques to probe two specific sites in transmembrane helices 4 and 6 of full length MsbA embedded in lipid bilayers. This probed the chemical environment at the local, site-resolved secondary structure level, and cryogenic conditions allowed the detection of conformational states that would be averaged out by molecular motions at higher temperatures.

The data clearly showed significant differences in chemical shift for both sites in TM4 and TM6, between the unliganded (apo) state and the vanadate (Vi)-trapped state. Vanadate replaces the cleaved γ-phosphate in the ADP+Pi-bound occluded NBS, and thereby mimics and locks the transition state of the hydrolysis reaction. These differences in chemical shift indicated changes in local environment and conformation between the two states for the monitored locations.

Notably, for the arginine residue R183 in TM4, in the apo state, the amide spectrum featured a shoulder at the chemical shift of the Vi-trapped state. Since the Vi-trapped state is expected to involve a closed NBD dimer, corresponding to the ATP-bound OF state, this was interpreted as indicating that the apo state involves a conformationally heterogeneous population of symmetrical MsbA dimers, comprising predominantly IF conformers with a sub-population of OF conformers. Thus, to explain these data, the Switch Model (SM) requires that the apo state samples the outward-facing closed-NBD dimer state, which is fundamentally at odds with the central principle that ATP binding is required to induce the transition to the OF (Figure 2).

In contrast, these NMR data can be interpreted naturally from the perspective of the Reciprocating Model (RM) as reflecting the functional asymmetry of the MsbA homodimer. Considering the scheme in Figure 4, asymmetry clearly always exists within the TMD dimer. Thus, for each of the probed sites in TMs 4 and 6, two sites in asymmetric conformation always exist within the MsbA homodimer. Hence, the observation that for most states analyzed by Spadaccini et al. (2018) [36], a significant shoulder or peak in the amide spectrum is observed at the chemical shift of both the Vi-trapped and apo states, simply reflects this asymmetry, indicating different conformations/environments of the two equivalent probed sites, that coexist within a single asymmetric MsbA homodimer.

Another significant feature of these NMR data can be illustrated by considering the spectra for R183 in TM4 in the apo and Vi-trapped states, in the presence of the drug ligand daunorubicin (Figure 3 in [36]). Thus, for the apo state, the area under the spectral curve approaches roughly twice that for the Vi-trapped state, indicating that in the transition from apo to Vi-trapped in the presence of daunorubicin, a large proportion of the monitored sites disappear from the signal. This pattern, whereby the Vi-trapped, drug-bound state displays a significantly smaller proportion of the sites detected in the apo state is also borne out by the data generally [36]. In addition, for the A314-C315 pair in TM6, in the transition from the apo to Vi-trapped states, only about half of the sites change to the Vi-trapped chemical shift, with the remainder staying at the apo chemical shift or disappearing.

Clearly, these findings are not consistent with a conformationally heterogeneous population of symmetrical MsbA dimers transitioning to a single more restricted and predominant conformation, in which case an increase in signal intensity would occur. However, from the perspective of the RM, it can be postulated that when one site is in a state whereby it is most restricted conformationally, such as in the substrate-bound, Vi-trapped state, the opposite equivalent site within the MsbA homodimer is disordered or conformationally heterogeneous such that it produces no significant peaks in the spectra.

A final notable feature of these data is the observation that the presence of the transportable ligand substrate Hoechst 33342 does not affect the putative equilibrium between IF and OF conformations in the apo state, and nor does it significantly affect the proportion of MsbA dimers that reach the Vi-trapped state. Since Hoechst 33342 was present at optimal concentration for stimulation of ATPase activity [37], these findings appear at odds with data for P-gp, which showed that substrates stimulated the formation of the Vi-trapped state [38] and that the extent of Vi-trapping was correlated with the fold stimulation of steady-state ATPase activity by the substrate [39].

From the perspective of the Switch model, presumably, substrate binding would lower the energetic barrier to the IF to OF transition, and be required to avoid uncoupled, futile cycling. Thus, if the IF to OF transition can occur even in the apo state, as required to explain the data, it appears difficult to explain how the substrate, either alone or in the presence of ATP and vanadate, does not affect this equilibrium.

Clouser and Atkins (2022) [40] reported hydrogen-deuterium exchange mass spectrometry (HDX) analysis of murine P-gp (MDR1A) in lipid nanodiscs. This study examined the differences in HDX between the nucleotide-free, pre-hydrolysis, and post-hydrolysis states, and the effect of drug ligands on these states, to characterize functional dynamic changes. Pepsin digestion and mass spectrometry to assess the identity and deuteration of the fragments was used to resolve dynamic changes to the level of, on average, approximately 10 amino acid segments.

The salient feature of the HDX coverage of P-gp, as allowed by localized accessibility to pepsin digestion, is a pronounced non-symmetric pattern of protection from pepsin cleavage with respect to the pseudo-symmetric structure of the P-gp molecule. While not immediately apparent when visualized on the apo IF structure [40], when visualized on the OF ATP-bound P-gp structure, it becomes clear that a region centered on the NBD2 helical domain, comprising a substantial part of the NBD2 helical domain, including the C-motif and downstream helix, and together with the entirety of ICLs 1 and 2, which are deployed around NBS1, are protected from pepsin digestion. In contrast, the broadly equivalent regions around NBS2 are not protected. Another notable asymmetry in this respect is that for NBD1, accessibility data for the D-loop was able to be obtained while that for the downstream D-helix was not, whereas for NBD2 the converse was true.

Moreover, while segments at the intracytoplasmic ends of TMs 10 and 11 were accessible to pepsin cleavage, neither the equivalent predicted membrane spanning regions of TMs 4 and 5, nor any other TM except for TM6, were accessible. While TM6 was accessible to pepsin cleavage up to a point on the extracellular side of the predicted midpoint of the membrane bilayer, the equivalent TM12 was completely inaccessible.

Presumably, the soluble regions of P-gp around NBS1 resistant to pepsin digestion comprised closely packed and/or buried regions of low dynamics. Thus, the soluble regions inaccessible to HDX analysis delineated elements of the conformation that remained constant in the study, regardless of ligands. In summary, the analysis found asymmetric dynamics of the NBD dimer and CHs whereby NBD1, CH4, and the NBD2 RecA-like domain are dynamic, whereas in contrast, CH1, CH2, and the NBD2 helical domain were not.

In the transition from nucleotide-free to pre-hydrolysis states in the absence of ligands, NBD1, the NBD2 RecA-like domain, a C-terminal segment of CH3, and CH4 undergo dynamical changes that render them less accessible. In the transition from the pre- to post-hydrolysis (Vi-trapped transition) states in the absence of ligands, there is a further decrease in accessibility of the NBD1 RecA-like domain, while there is a mixed pattern of changes in dynamics for the NBD1 helical domain and for the NBD2 RecA-like domain.

Thus, this study of murine P-gp delineates asymmetric and independent accessibility and dynamics of the RecA-like and helical domains of the NBDs as being involved in functional changes of the transport process. While these findings appear complex and indecipherable from the perspective of the Switch Model, they are in complete accord with our previous MD data and Constant Contact model [5], in which three distinct relative motions of the NBD major subdomains are integrated in an alternating site scheme of ATP hydrolysis [41].

Lacabanne et al. (2019: 2022) [42,43] used ^13^C and ^31^P solid-state NMR to probe conformational changes and dynamics during the catalytic cycle of the multidrug ABC transporter BmrA, reconstituted in lipids of the native organism Bacillus subtilis. Combined, the ^13^C and ^31^P NMR analyses clearly revealed an asymmetric structure of the NBD dimer in the BmrA:ADP:Vi transition-state mimic, in which one NBS was occupied by tightly bound ADP:Vi and is not accessible, while the second NBS could bind ADP, but not ATP, and was sufficiently open to allow exchange with nucleotides from the solvent.

The study also found that in the presence of ADP alone, both NBSs were in a state close to that of the apo state, and essentially indistinguishable from the accessible site in the ADP:Vi-trapped state. In addition, analysis of the BmrA:ADP state revealed two distinct binding modes for ADP, one in which the nucleotide was immobilized such that a strong signal was produced, and the other weaker and more transient.

Lacabanne et al. (2022) [43] proposed a variant of the Switch Model for the catalytic cycle based on their NMR data and the structures. Thus, BmrA transitions from the IF conformation with widely separated NBDs to the OF NBD sandwich dimer upon binding of ATP in both NBSs. Notably, however, in the BmrA:ADP state, the NBSs were found to be in the same state as the accessible site in the ADP:Vi-trapped state, which, since the opposite, closed NBS is only some 20Å distant, must be semi-closed, with both halves of the NBS in close proximity.

Thus, the NMR data rather are consistent with the BmrA:ADP state, and thus the apo state, being one in which the NBDs are in relatively close apposition. Therefore, the NMR data do not clearly show or support widely separated NBDs, and this interpretation relied on, and deferred to, the authors’ previous data derived from detergent-solubilized BmrA, such as cryo-EM and X-ray structural and small-angle neutron scattering analyses [44,45].

Furthermore, while the authors concluded that the apo and BmrA:ADP states are symmetrical, this was based on the symmetry of the residues providing an above threshold signal. This did not preclude an asymmetric NBD dimer wherein equivalent regions from each NBD undergo anticorrelated motions, such that they produce sub-threshold signals. Indeed, the finding of two distinct affinity binding sites for ADP in the BmrA:ADP state appears difficult to reconcile with a symmetrical IF conformation (Figure 2) but is consistent with an asymmetric NBD dimer in which the NBS alternately open and close.

Of relevance to this question are studies using the reagent NBD-Cl, that can form a covalent bond with cysteine residues, which occur in the P-loop in both NBSs of Pgp [46,47]. In the absence of nucleotides, this reaction occurs at a stoichiometry of 1 mol of NBD-Cl/mol of Pgp; the reaction of one NBS with NBD-Cl severely impedes the reaction of reagent with the other. This is clearly consistent with an asymmetric NBD dimer with one closed and one open NBS in the apo state. It is clearly not consistent with the wide apart NBDs in apo structures of ABCB1 homologues, wherein both P-loops are in the same conformation and are clearly accessible.

A final point for the assessment of the Switch and Reciprocating Models is to note that analysis from the perspective of the Switch Model typically involves extreme and artificial states, such as the absence of all ligands (apo), or saturating concentrations a single species such as a substrate, ADP, or ATP analogue. In contrast, from the perspective of the RM, ABC transporters can be analyzed and understood using, and in terms of, physiological concentrations of all relevant ligands.

## 3. Summary and Conclusions

Here, we have discussed data from recent high-level analyses of ABC MDR exporters, reconstituted in lipid membranes, from the perspective of the Switch and Reciprocating Models for the function of ABC transporters. The salient feature of these data overall is how starkly and fundamentally they are at odds with the symmetrical nature of ABCB1 structures and its implications, as embodied in the Switch Model.

For the Spadaccini et al. (2018) NMR analysis of MsbA [36], the perspective of the SM requires that the data indicate a significant population of OF MsbA dimers in the absence of ligands and requires ignoring the absence of increased peak intensity at the Vi-trapped chemical shift, relative to the apo state, in the presence of a trapped nucleotide, expected from this explanation. What is also problematic from this perspective is the observation that the presence of the transportable ligand Hoechst 33342 does not affect the putative equilibrium between IF and OF conformations in the apo state, and nor does it affect the proportion of MsbA dimers that reach the Vi-trapped state; in other words, the data gives no indication that the substrate affects the transition from IF to OF.

In contrast, the seemingly contradictory and inexplicable nature of these findings dissolves when viewed from the perspective of the RM, where they can be broadly understood in terms of the asymmetry of the MsbA homodimer.

The Clouser and Atkins (2022) HDX analysis of murine P-gp [40] found asymmetry of the HDX coverage, as allowed by pepsin digestion, whereby some putative TM segments, and a subset of the soluble domains, are accessible while their pseudo-symmetrical structural counterparts are not. In addition, the analysis delineated the RecA-like and helical subdomains as undergoing independent functional dynamic changes and found generally asymmetry of the dynamics observed between all states analyzed.

From the perspective of the Switch Model and the structures, these HDX data appear inconsistent or complex and indecipherable; most significantly, they are not enlightening in any clear, coherent, and meaningful way. In terms of the RM, these HDX data are broadly consistent and intelligible, and this study may, therefore, be viewed as informative and revealing. Moreover, they further support the notion that the structures should be regarded with greater circumspection, in particular, their complete inability to account for the extensive accessibility of TM6 to pepsin digestion.

The Lacabanne et al. 2022 solid-state NMR study of BmrA [43] confirmed and elaborated on previous findings of an asymmetric NBD dimer in the ADP+Vi trapped transition state [46,48,49]. This analysis also found that in the apo and ADP-bound states, the NBSs were indistinguishable from the accessible NBS in the Vi-trapped state, which, due to the geometry of the NBDs, must involve closely apposing RecA-like and helical domains from opposite NBDs. Thus, these data do not support the wide apart NBDs predicted and required by the Switch Model. Moreover, neither the structures nor the Switch Model can explain how and why the ADP:BmrA state exhibits two different binding affinities for ADP, or how and why the accessible site in the ADP:Vi-trapped state cannot bind ATP from the solvent. In contrast, these asymmetries are clearly integral to the RM/CC model.

In the scheme proposed by Lacabanne et al. (2022) [43], the transporter progresses from the ATP-bound NBD sandwich dimer to the ADP:Vi-trapped state via the hydrolysis of the two bound ATP molecules, with the ADP:Vi-trapped state forming after hydrolysis of the second ATP and replacement of the released phosphate with vanadate. This raises a central question touched on by the authors: Does destabilization and dissociation of the NBD sandwich dimer require the hydrolysis of both bound ATPs, or only one? From an equilibrium energetics perspective, if it takes two ATPs to stabilize the closed NBD dimer, then hydrolysis of one should be enough to cause NBD dissociation. This idea, however, appears inconsistent with the NMR findings here, where the NBDs are shown to be stably held together by only a single ADP:Vi complex bound in one NBS.

A further point of relevance is the conclusion that the NMR data for BmrA, together with a body of other enzymological data for BmrA and its homologues, support a stoichiometry of one bound ADP:Vi per transport complex, within an asymmetric NBD dimer [46,48,49]. This is clearly at odds with 3D structures of the maltose transporter and the BmrA homologue MsbA, solved in the presence of ADP:Vi, which show two ADP:Vi bound in a symmetric NBD sandwich dimer [34,50,51]. Indeed, the double Vi-trapped state has never been observed other than in structural studies. The question thus arises: If the symmetric structures of the ADP:Vi-bound NBD dimer for MsbA is an artefact, what is the basis for confidence that other symmetrical NBD sandwich dimers, observed in structures of ABCB1 homologues, and which are crucial evidence supporting the Switch Model, represent a physiological state?

We have argued that the data discussed herein, while of the highest quality, are at best inconclusive as to whether they support the Switch over the Reciprocating Model. For the most part, the data are at odds with the Switch Model and/or the structures and are far more naturally and readily accommodated by the Reciprocating Model. Crucially, none of these data were able to be interpreted and understood in a way that further articulated and illuminated the Switch mechanism, with the exception of the Lacabanne et al. studies [42,43], where a hybrid of the Switch and CC/RM models was proposed.

Despite the significant problems considered in foregoing discussions of recent NMR and HDX studies, for the most part, investigators appear undeterred in interpreting their data from the perspective of the structures and the Switch mechanism, and in many instances simply left unremarked salient aspects of their results that were clearly unable to be reconciled from this perspective. Thus, the general adherence to the structures is not commensurate with their success in explaining the data and illuminating the molecular mechanism. It appears that the strength of this adherence is founded on the notion that the structures are very unlikely to be wrong.

## Figures and Tables

**Figure 1 ijms-24-02624-f001:**
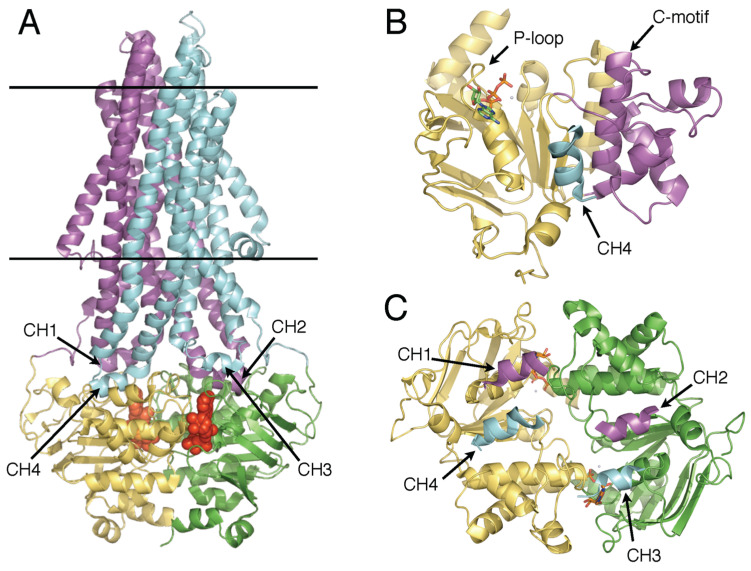
Structure of ABCB1 ABC Exporters. The structure of human P-glycoprotein in the ATP-bound, outward facing conformation (PDB ID: 6C0V) is illustrated in cartoon representation. (**A**) View from the plane of the membrane, approximately along the plane of the NBD dimer interface, with predicted limits of the membrane shown in black horizontal bars. Atoms of MgATP molecules shown as red spheres. TMD1 magenta, TMD2 cyan, NBD1 yellow, NBD2 green. Coupling helices of the TMDs that interface with the NBDs are indicated. (**B**) NBD1 with RecA-like domain yellow, helical domain magenta, CH4 cyan. ATP shown in stick form with oxygen red, nitrogen blue, carbon green, phosphorous gold. (**C**) NBD sandwich dimer with NBDs and CHs colored as in (**A**), with ATP molecules bound at the interface represented as in (**B**).

**Figure 2 ijms-24-02624-f002:**
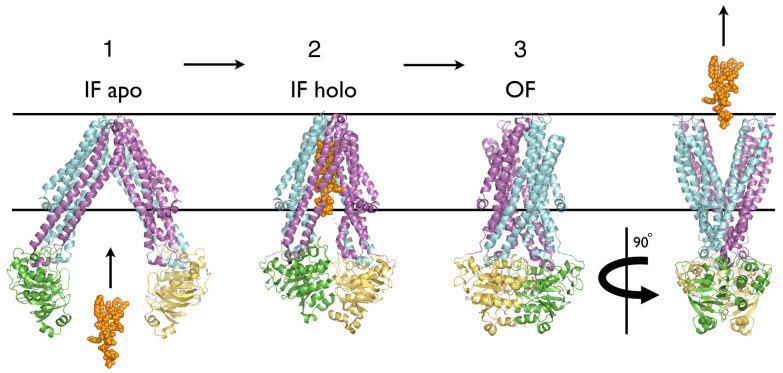
Switch Model for ABC Exporters. Alternating access Switch Model for the function of an ABC exporter. Structures of orthologues of the bacterial lipid A exporter MsbA. (i) PDB 6BL6 [32] *Salmonella typhimurium*; (ii) PDB 6BPL [33] *E. coli*; and (iii) PDB 3B6O [34] *Salmonella typhimurium*. MsbA is shown in cartoon ribbon representation with one protomer colored magenta and yellow and the other cyan and green for the TMD and NBD, respectively. Parallel lines represent the extracellular (upper) and intracellular (lower) membrane boundaries. Lipid A ligand is shown in atomic sphere representation and colored orange, ATP in is stick form. (**Stage 1**) Unliganded transporter in inward-facing resting state. (**Stage 2**) Lipid A ligand binds to inward-facing TMDs. (**Stage 3**) ATP binding to each NBD induces tight dimerization of NBDs and transition of TMDs from inward- to outward-facing conformation, as the substrate is released to the extracellular space. Two orthogonal views of (**Stage 3**) are shown to illustrate both NBD dimerization and outward facing conformation of TMDs.

**Figure 3 ijms-24-02624-f003:**
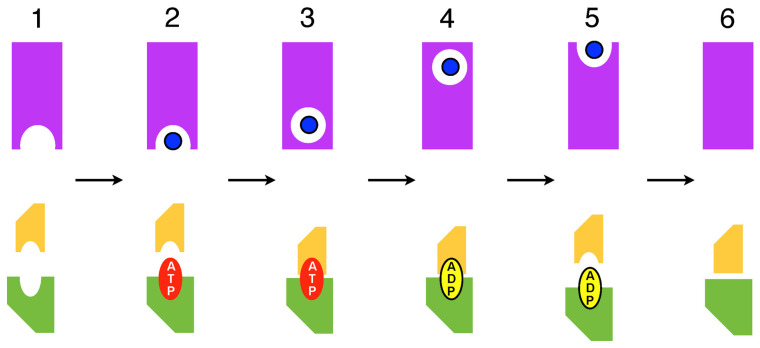
Coupling between the ATP hydrolysis and substrate translocation cycles in the Reciprocating Model. Scheme for an ABC exporter is shown. The cycle of ATP binding, occlusion, and hydrolysis, followed by NBS opening and product release in one NBS is directly coupled to an analogous cycle of substrate binding, occlusion, translocation, and presentation at the extracellular side and release in one of two functionally separate substrate translocation pathways. Top Panel: A single TM translocation channel (TMC) is depicted as a magenta rectangle with the substrate-binding site as an inward- or outward-facing semicircle. The occluded substrate-binding site is depicted as a circle within the TMC and the substrate as a blue dot. Bottom Panel: One composite active site (NBS) is depicted by two shapes representing the helical domain of one NBD (smaller, yellow) and the RecA-like domain of the opposite NBD (larger, green). The numbered steps that define the cycle are: (**1**) High affinity inward-facing substrate binding site is empty and NBS active site is open and empty. (**2**) Substrate binds to the inward-facing site and ATP binds to NBS. (**3**) Substrate is sequestered in an occluded site on the cytoplasmic side of the membrane; ATP is occluded in the active site. (**4**) ATP is hydrolyzed, coupled with substrate translocation to an occluded site at the extracellular side of the membrane. (**5**) Opening of ADP-bound active site corresponds to the substrate bound to extracellular-facing low affinity site. (**6**) Substrate and ADP nucleotide disengage; NBS and TMC are in a low affinity states.

**Figure 4 ijms-24-02624-f004:**
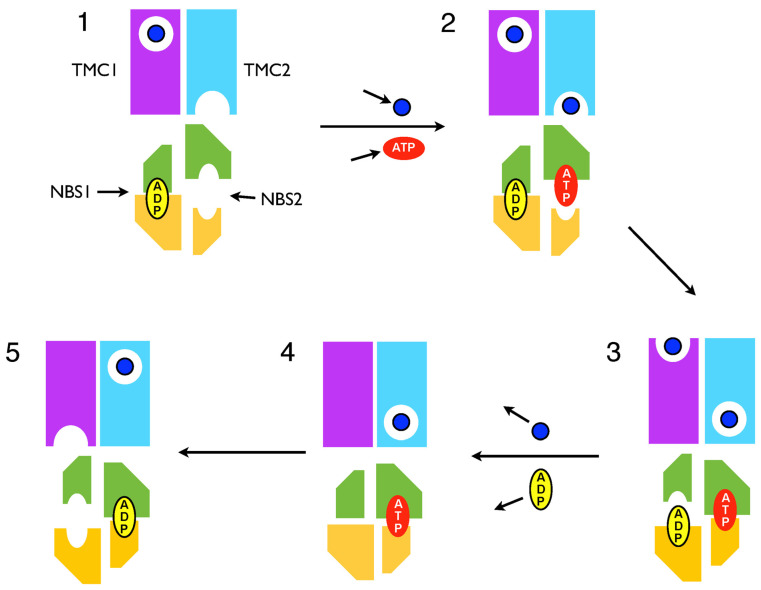
Reciprocating Model for an ABC Exporter. The translocation cycle of the full transporter involves two separate coupled ATP hydrolysis and substrate translocation cycles, which function 180 degrees out of phase (the cycle of a single TMC/NBS pair is depicted in Figure 3). One half cycle is shown in this figure. The TMD channels are depicted as magenta or cyan rectangles and referred to as TMCs 1 and 2, respectively. NBDs 1 and 2 are colored yellow and green, respectively. The substrate is depicted as a blue dot, ATP as a red lozenge, and ADP as a yellow lozenge. It is important to note that from a structural perspective, while the NBDs are depicted specifically, in contrast, the illustrated TMCs do not correspond to each TMD, rather the TMD dimer is imagined to embody two functionally distinct channels in a structurally unspecified way. The cycling scheme is as follows: (**1**) The exporter is poised to bind the substrate to the high affinity inward-facing site of TMC2 and ATP to the open high affinity NBS2. Occluded transported substrate and ADP await dissociation from TMC1 and NBS1, respectively. (**2**) Substrate binds to TMC2 and ATP binds to NBS2. (**3**) Binding of substrate and ATP induces an isomerization of the transporter whereby NBS2 closes and occludes ATP, while ADP-bound NBS1 opens. Coupled with this, the substrate bound to the inward-facing site of TMC2 becomes occluded and the substrate is presented at the extracellular side of TMC1. (**4**) Following ADP release from NBS1 and substrate release from TMC1, the transporter isomerizes to an occluded conformation in which empty NBS1 is closed. The transporter is now competent for ATP hydrolysis. This is close to the conformation captured by vanadate trapping. (**5**) Upon ATP hydrolysis, the substrate is translocated to an occluded site on the extracellular side of TMC2, TMC1 opens a high affinity inward-facing substrate binding site, and NBS1 opens to present a high affinity ATP binding site. This stage is equivalent to that depicted in Stage 1, which is for the opposite side of the cycle.

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
