# Peer review of "The Switch and Reciprocating Models for the Function of ABC Multidrug Exporters: Perspectives on Recent Research"

_ijms, 2023, doi:10.3390/ijms24032624_

Round 1

Reviewer 1 Report

The manuscript is well written and highlights the recent advancements in better understanding the mechanism of ABC protein transport cycle. The reviewer has no further comments for any revision. 

Author Response

MINOR REVISIONS

REVIEWER 1

Accept as it is.

We thank the reviewer.

Reviewer 2 Report

In this review article, Jones and George compare the available structures of P-glycoprotein, two of its homologs, MsbA and BmrA, to two suggested transport models: the Switch or Alternating Access Model and the Reciprocating Model. The authors list several studies whose findings are better explained by the Reciprocating model than the Switch model. Based on these explanations, the Reciprocal model seems to be a better fit. Still, it might need more papers and reviews to convince all structural biologists that the Reciprocal model is more suitable to the available data. This review might help to get there.

Minor: 

I would add “active” in lines 7 and 32 to read “primary active transporters” rather than “primary transporters”.

In line 108, “for any given ABC remains elusive” should be “for any given ABC transporter remains elusive”.

In line 176, the authors talk about three ABCB1 ABC exporters. I understand that only P-glycoprotein is called ABCB1 (with the gene symbol of ABCB1), while the bacterial transporters have different names. Why not change the sentence to “analyses of ABCB1 and two of its bacterial homologs” or something like this?

In line 247 “mdra1” should be “MDR1A” based on the current recommendations of the genome nomenclature committees.

Author Response

MINOR REVISIONS

REVIEWER 2

In this review article, Jones and George compare the available structures of P-glycoprotein, two of its homologs, MsbA and BmrA, to two suggested transport models: the Switch or Alternating Access Model and the Reciprocating Model. The authors list several studies whose findings are better explained by the Reciprocating model than the Switch model. Based on these explanations, the Reciprocal model seems to be a better fit. Still, it might need more papers and reviews to convince all structural biologists that the Reciprocal model is more suitable to the available data. This review might help to get there.

Minor:

I would add “active” in lines 7 and 32 to read “primary active transporters” rather than “primary transporters”.

In line 108, “for any given ABC remains elusive” should be “for any given ABC transporter remains elusive”.

In line 176, the authors talk about three ABCB1 ABC exporters. I understand that only P-glycoprotein is called ABCB1 (with the gene symbol of ABCB1), while the bacterial transporters have different names. Why not change the sentence to “analyses of ABCB1 and two of its bacterial homologs” or something like this?

In line 247 “mdra1” should be “MDR1A” based on the current recommendations of the genome nomenclature committees.

All of this reviewer’s suggested changes have been made in the revised manuscript. We thank the reviewer.

Reviewer 3 Report

Peter M. Jones and Anthony M. George describes a novel mechanism for ABC transporter functions which has been proposed known as the Switch or Alternating Access Model. The review is very interesting and complete, offering an update in ABC transports functions.

I suggest that the study is suitable for publication, thus I suggest only minor corrections before potential publication of manuscript.

1. I consider that the proposals and analyses that the authors are explained clearly and novel with respect to the information that exists in the literature so far. 

2. The wording is understandable and correct.

3. The content of each topics of each section, was written, explained and detailed with clarity.

4. The figures are explicit and of very good quality.

5. Figure captions reflect what corresponds to the figure.

Minor.

It should be interesting to consider adding a new section  about the multidrug resistance phenotype by the ABC transporter, mainly in cancer.

Author Response

MINOR REVISIONS

REVIEWER 3

Peter M. Jones and Anthony M. George describes a novel mechanism for ABC transporter functions which has been proposed known as the Switch or Alternating Access Model. The review is very interesting and complete, offering an update in ABC transports functions.

I suggest that the study is suitable for publication, thus I suggest only minor corrections before potential publication of manuscript.

  1. I consider that the proposals and analyses that the authors are explained clearly and novel with respect to the information that exists in the literature so far.
  2. The wording is understandable and correct.
  3. The content of each topics of each section, was written, explained and detailed with clarity.
  4. The figures are explicit and of very good quality.
  5. Figure captions reflect what corresponds to the figure.

Minor.

It should be interesting to consider adding a new section about the multidrug resistance phenotype by the ABC transporter, mainly in cancer.

We have considered the reviewer’s minor suggestion and believe that a new section on ABC transporters and cancer would be more suited to a broader review. To include it in this manuscript would dilute the emphasis of the comparisons we make with different models. We thank the reviewer.